# Integration of mRNA-miRNA Reveals the Possible Role of *PyCYCD3* in Increasing Branches Through Bud-Notching in Pear (*Pyrus bretschneideri* Rehd.)

**DOI:** 10.3390/plants13202928

**Published:** 2024-10-18

**Authors:** Ze-Shan An, Cun-Wu Zuo, Juan Mao, Zong-Huan Ma, Wen-Fang Li, Bai-Hong Chen

**Affiliations:** College of Horticulture, Gansu Agricultural University, Lanzhou 730070, Chinamazohu@163.com (Z.-H.M.);

**Keywords:** pear, bud-notching, branching, RNA-seq, hormone signaling, *PyCYCD3*

## Abstract

Bud-notching in pear varieties with weak-branches enhances branch development, hormone distribution, and germination, promoting healthier growth and improving early yield. To examine the regulatory mechanisms of endogenous hormones on lateral bud germination in *Pyrus* spp. (cv. ‘Huangguan’) (*Pyrus bretschneideri* Rehd.), juvenile buds were collected from 2-year-old pear trees. Then, a comprehensive study, including assessments of endogenous hormones, germination and branching rates, RNA-seq analysis, and gene function analysis in these lateral buds was conducted. The results showed that there was no significant difference in germination rate between the control and bud-notching pear trees, but the long branch rate was significantly increased in bud-notching pear trees compared to the control (*p* < 0.05). After bud-notching, there was a remarkable increase in IAA and BR levels in the pruned section of shoots, specifically by 141% and 93%, respectively. However, the content of ABA in the lateral buds after bud-notching was not significantly different from the control. Based on RNA-seq analysis, a notable proportion of the differentially expressed genes (DEGs) were linked to the plant hormone signal transduction pathway. Notably, the brassinosteroid signaling pathway seemed to have the closest connection with the branching ability of pear with the related genes encoding *BRI1* and *CYCD3*, which showed significant differences between lateral buds. Finally, the heterologous expression of *PyCYCD3* has a positive regulatory effect on the increased *Arabidopsis* growth and branching numbers. Therefore, the *PyCYCD3* was identified as an up-regulated gene that is induced via brassinosteroid (BR) and could act as a conduit, transforming bud-notching cues into proliferative signals, thereby governing lateral branching mechanisms in pear trees.

## 1. Introduction

The pear (*Pyrus bretschneideri* Rehd.) is recognized as one of the most significant fruit crops globally due to its abundant nutrition, high output, and substantial financial value [1]. To enhance yield and optimize profitability, the utilization of well-branched fruit trees in production is crucial [2,3]. However, the apparent apical dominance of fruit trees, the low germination rate of the lower branches, and the excess growth of the top branches have become one of the problems to be solved in the structure of fruit trees [4,5,6,7,8,9]. The bud-notching technique has been increasingly used to promote the sprouting of auxiliary branches and backbone branches of apple and pear fruit trees. Previous studies on apple and pear show that bud-notching can improve the germination rate and growth, branch strength, increase the growth amounts of branch, adjust the composition of branches, and then achieve the purpose of increasing production [10]. Interestingly, apical dominance has long been considered to be orchestrated by hormone and sugar-dependent mechanisms [11,12]. The hormonal regulatory network that controls shoot branching has been extensively studied in *Arabidopsis* and pea (*Pisum sativum*) [13,14,15]. Nevertheless, the hormonal signaling mechanism involved in promoting lateral bud growth through bud-notching in pear trees remains unclear at this stage.

Physiological studies indicate that the influence of external factors on plant growth and development is closely related to the change in endogenous hormones in plants. This is because hormonal changes influence nutrient transport and distribution within plants [16,17]. Endogenous hormones play an important role in the regulation of fruit trees at all stages of the growth cycle, including sprouting, bud differentiation, and leaf dormancy [18]. Research has suggested that indole acetic acid (IAA), gibberellin (GA), and zeatin riboside (ZR) can promote bud germination and plant growth, while abscisic acid (ABA) can promote bud dormancy and bud differentiation, belonging to the group of growth-inhibiting hormones [19]. Additionally, levels of endogenous hormones such as GA_3_, ABA, ZR, and cytokinin (CTK) in buds treated with bud-notching have shown significant changes [20,21,22]. Specifically, the application of CTK at a specific concentration in the spring prior to apple tree sprouting can effectively trigger the growth of dormant buds [23]. The necessity of type-A ARRs for CK-mediated bud development might elucidate the increased expression of *MsARR3/5* observed during bud development [24]. The auxin signaling pathway plays a significant role in influencing cell growth, with *Aux/IAA* and the transcription factor *ARF* responding to auxin to regulate stem development through the control of cell division [25]. The suppression of abscisic acid (ABA) and subsequent activation of *DREB/CBFs* and *GA2OX* family members, in addition to the up-regulation of *GIDs*, GA biosynthesis genes, and the induction of genes positively regulated by GA, indicate that GA-mediated signaling likely contributes to the induction of growth [26]. In *Arabidopsis*, *AtBRC1* was observed to bind to the *AtNCED3* promoter and positively regulate abscisic acid (ABA) biosynthesis to suppress axillary bud outgrowth [27]. Evidence has identified brassinosteroid (BR) signaling as a component of axillary meristem formation [28]. In addition, downstream BR signaling factors have been shown to be shared by distinct signaling pathways, controlling different aspects of growth and development, including cell elongation, shoot branching, stomata formation, lateral root formation, and xylem differentiation [29]. The BR signal transduction pathway comprises numerous components, such as *BRI1*, *BAK1*, *BKI1*, *BSK1*, *CDG1*, *BSU1*, *BIN2*, and *BZR1/2* [30,31,32,33,34,35]. A study demonstrated that BR induces *CYCD3* transcription and promotes cell division, and that *CYCD3* also mediates the BR signal [36]. In addition, in BR-treated apple trees, the expression levels of cell-growth related proteins *MdCYCD1;1*, *MdCYCD3;2*, and *MdCYCB3;1* were significantly elevated compared to the control trees [37,38,39,40]. The up-regulation of cell proliferation genes such as *MsCYCD3* was observed at stage “1” in MB, while *MsPCNA2* was down-regulated compared to stage “0” in MB. Notably, *AtCYCD3* was shown to be essential for cytokinin-mediated functions, in particular the regeneration of shoots from callus [41].

This study highlights the significance of bud-notching technology for enhancing shoot branching in pear trees. Although some previous work has been conducted on pear budburst, the mechanisms of how its pear budburst-related hormones and associated genes are regulated remain unclear. We investigated germination and branching rates, endogenous hormones, transcriptome analysis (RNA-seq), and qRT-PCR to gain insights into this process. Specifically, we analyzed transcriptomic differences in lateral buds before outgrowth in the S0, S1, and S2 of ‘Huangguan’ pear, which exhibits increased branching due to bud-notching during its growth cycle. The findings of this study suggest that the transport of auxin and BR from lateral branches to the stem may be crucial for the development of lateral buds in pear trees, with *PyCYCD3* potentially serving as a key regulator in bud germination and branch growth. Based on the present results, we hypothesize that the up-regulation of the *PyCYCD3* gene might be an important modulator of auxin and BR signaling during pear branching. The primary research framework of this study is illustrated in Figure 1.

## 2. Results

### 2.1. Samples Treatment and Collection

All ‘Huangguan’ pear (*Pyrus bretschneideri* Rehd.) trees were cultivated in the same orchard under uniform management practices and sharing an identical genetic background. Prior to budding in late March, pear trees with similar growth patterns and heights were carefully selected. Bud-notching was performed by making epidermal incisions 30 cm below the treetop, positioned 1 cm above each bud. These incisions extended on both sides of the bud, covering approximately one-third of the stem’s circumference. Lateral bud samples were collected 15 days post-notching. The first bud sample (S0), which was collected from a position 30 cm below the apical bud, demonstrates an inability to naturally promote lateral branching (Figure 2b). The second bud sample (S1), representing the first lateral bud directly below the apical bud, demonstrates a natural ability to promote lateral branching (Figure 2b). The third bud sample (S2) was collected from the first bud that underwent bud-notching 30 cm below the apical bud, and naturally exhibited branching (Figure 2a). From each position, ten bud samples were gathered, pooled collectively, and the entire procedure was reiterated thrice. An elaboration of the comprehensive sample collection process is depicted in Figure 2.

### 2.2. Analyzing Germination Rate, Branching Rate, and Endogenous Hormone Levels in Lateral Buds

The germination rate of pear trees by bud-notching treatment was subjected to statistical analysis. The sprouts of pear trees treated with bud-notching predominantly emerged within a 60 cm radius from the apical bud. The germination rate in the bud-notching group reached 90.01%, while the control group exhibited a germination rate of 89.53%, and there was no significant difference in germination rates between the two groups (Figure 3a, Appendix A). The application of bud-notching treatment significantly enhanced the sprout rate of pear trees, as evidenced by the results presented in Figure 3b and Appendix A. Specifically, the average growth promotion rate of long branches of pear trees in the bud-notching group (91.21%) was significantly higher than that of the control group (33.63%) (*p* < 0.01). Additionally, there was no significant difference in the average promotion rate of medium branches between the control group (4.16%) and the bud-notching group (4.18%) (*p* < 0.05). The mean rate of short shoot promotion in the control group was 77.60%, compared to 20.01% in the bud-notching group. The promotion rate of short branches in the control group was found to be significantly higher than that in the treatment group (*p* < 0.01) (Figure 3b). The comparison between the bud-notching and control groups indicated that bud-notching significantly enhanced the growth of pear trees at the initiating side buds.

The peak area values of five hormones were quantified using liquid chromatography, and the concentrations of the hormones were subsequently determined (Figure 3c, Appendix A). The findings indicated that the levels of ZR, GA_3_, IAA, and BR in S1 and S2 were significantly elevated compared to those in S0, while the concentration of ABA did not exhibit a significant difference among S0 and S2. However, the levels of ABA in S1 were significantly elevated compared with S0 (*p* < 0.05). The ratio of BR/ABA in S1 was found to be statistically lower than that in S0 (*p* < 0.05), while the BR/ABA ratio in S2 was significantly higher than that in both S1 and S0 (*p* < 0.05) (Appendix A). It is hypothesized that the up-regulation of growth-promoting hormones such as ZR, GA_3_, IAA, and BR, along with the maintenance of ABA at a stable level, may positively regulate the initiation of lateral branches following bud-notching.

### 2.3. Analysis of Sequencing Data and DEGs

To better understand the molecular mechanism of shoot branching by bud-notching in ‘Huangguan’ pears, we conducted a comparative transcriptomic analysis on new young shoots branching between S0, S1, and S2 groups. After quality control, 65.15 Gb of clean data were obtained, with average quality Q30 scores of >93.96% for each library, respectively, and an average GC content of 44.64%. For the three sample libraries, 68,728,487, 73,564,496, and 751,55,150 reads were mapped to the pear genome, representing 64.24–66.49% of the clean reads from the three samples (Appendix A).

A total of 2154 DEGs were detected between S0 and S1, of which 1131 genes were up-regulated and 1023 genes were down-regulated. Similarly, 3036 DEGs were detected between S0 and S2, where 1339 genes were up-regulated and 1697 genes were down-regulated. Of these, 492 genes were found to be differentially expressed at both two confrontations (C1: S0_vs_S1, C2: S0_vs_S2) (Figure 4a). The comprehensive expression profile of these 492 DEGs was presented in Figure 4a, with 238 genes up-regulated and 254 genes down-regulated (Figure 4b, Appendix A). Clean reads with lengths from 18 to 30 bp were obtained from S0, S1, S2. The percentage of bases Q30 reached at least 98.52% of all the samples. In addition, 505 miRNAs were identified, including 141 known and 364 new miRNAs (Appendix A). Examination of miRNA expression in three samples identified 149 and 39 differentially expressed miRNAs (DEGs) at two confrontations, C1 and C2, respectively. Out of the total, 130 genes were found to be up-regulated and 19 genes were down-regulated in C1. Additionally, 27 genes were up-regulated and 12 genes were down-regulated in C2. In both confrontations, C1 and C2, nine differentially expressed genes (DEGs) exhibited up-regulation, while two DEGs displayed down-regulation (Figure 4d). Specifically, the expression profile of often differentially regulated miRNAs is presented in Figure 4c.

Functional classification of both mRNA and miRNA was conducted using the KEGG database (Figure 4e,f). For this analysis, 22 significant KEGG pathways were selected. The mRNA KEGG pathway analysis revealed the following distribution of DEGs: there were 25 and 32 DEGs from ‘Cellular Processes’, 45 and 35 DEGs from ‘Environmental Information Processing’, 53 and 86 DEGs from ‘Genetic Information Processing’, 110 and 155 DEGs from ‘Metabolism’, along with 21 and 24 DEGs from ‘Organismal Systems’ in C1 and C2, respectively (Figure 4e, Appendix A). In miRNA KEGG pathway analysis, there were 7 and 10 DEGs from ‘Cellular Processes’, 13 and 11 DEGs from ‘Environmental Information Processing’, 66 and 55 DEGs from ‘Genetic Information Processing’, 60 and 35 DEGs from ‘Metabolism’, along with 13 and 8 DEGs from ‘Organismal Systems’ in C1 and C2, respectively (Figure 4f, Appendix A). ‘Plant hormone signal transduction’, ‘Protein processing in endoplasmic reticulum’, and ‘Phenylpropanoid biosynthesis’ were the top three pathways with the largest number of associated DEGs in C1 and C2 (Figure 4e). A significant number of DEGs in C1 and C2 were related to ‘plant hormone signal transduction’, with 43 and 31 genes in the mRNA KEGG pathway and 9 and 6 genes in the miRNA KEGG pathway, respectively.

### 2.4. Plant Hormone Signaling Pathway Analysis of mRNA DEGs

Based on the KEGG pathway analysis, the ‘plant hormone signaling’ pathway was found to be essential for this research. Consequently, an in-depth analysis using the mRNA DEGs associated with this pathway was conducted. Within the auxin signaling transduction pathway, genes such as *AUX1*, *AUX/IAA*, *SUAR*, *ARF*, *GH3*, *PP2C*, *ABF*, *BRI1*, and *CYCD3* were enriched (Figure 5a, Appendix A). Most of the predicted auxin-induced protein genes (gene38584, 38585, 38587, 17817, 27027, newgene 29561) belonging to the *SUAR* family were identified to be down-regulated in both S1 and S2 when compared with S0. *ARF* (Pyrus_newGene_32024) was identified to be up-regulated in both S1 and S2 when compared with S0. *GH3* (gene11049) was identified to be down-regulated in both S1 and S2 when compared with S0, *GH3* (gene1810, gene34115, gene34117) was identified as being up-regulated in both S1 and S2 when compared with S0. Within the GA signaling transduction pathway, genes *GID1* and *TF* were found to be enriched, of which *GID1* (gene22380) and *TF* (gene28794) were identified to be down-regulated in both S1 and S2 when compared with S0. In the ABA signaling transduction pathway, *PYR/PYL*, *PP2C,* and *ABF* were enriched. Specifically, *PYR/PYL* (gene24413) was down-regulated in S1 and up-regulated in S2 when compared with S0, while *PP2C* (gene12403, gene24075, gene26342) was up-regulated in S1 and down-regulated in S2 when compared with S0. *ABF* (gene23064) was identified as being up-regulated in both S1 and S2 when compared with S0. Within the CTK signaling transduction pathway, *AHP* and *B-ARR* were enriched. Among these, *AHP* (gene22385) and *B-ARR* (gene9903) were identified as being up-regulated in S1 and down-regulated in S2 when compared with S0. Lastly, within the BR signaling transduction pathway, *BRI1*, *BSK,* and *CYCD3* were enriched, with *BRI1* (gene6361) and *CYCD3* (gene1164) identified as being down-regulated in S1 and up-regulated in S2 when compared with S0, while *BSK* (Pyrus_newGene_23435) was up-regulated in S1 and down-regulated in S2 when compared with S0.

### 2.5. Analysis of the Interaction Network of miRNAs and Target mRNAs

The identification of key genes associated with shoot branching following bud-notching in pears was achieved through the KEGG pathway and miRNA-mRNA analysis. Through an analysis of the regulatory mechanisms of miRNAs and their target genes, a significant number of miRNAs were identified as targeting seven genes associated with shoot branching in pear trees (Appendix A). After analysis, only one miRNA was identified to target at least two pear shoot branching-related genes, which was cpa-miR166e (*GH3* and *ABF*). Additionally, ten miRNAs were identified as targeting only one pear shoot branching-related gene: aly-miR157d-3p (*ABF*), cem-miR166g (*ABF*), atr-miR166b (*ABF*), gma-miR166m (*ABF*), bdi-miR166e-3p (*ABF*), hbr-miR166a (*ABF*), cpa-miR166e (*ABF*), aly-miR166a-3p (*ABF*), mes-miR166i (*ABF*), cis-miR166a (*ABF*). One miRNA was identified to target one pear shoot branching-related genes, which were cpa-miR166e (*GH3*), gma-miR390e (*SAUR*), gma-miR162a (*SAUR*), hbr-miR156 (*PP2C*), aly-miR390a-5p (*BRI1*), and cme-miR396e (*CYCD3*). In addition, a regulatory network of miRNA-mRNA related to pear shoot branching-related genes was constructed (Figure 5b, Appendix A). The results suggested that genes can negatively regulate the expression of miRNA. For example, ABF was up-regulated in S1, while the target ten miRNAs were identified as being down-regulated in S1; CYCD3 was up-regulated in S2, but the target miRNA (cme-miRNA396e) was identified as being down-regulated in S2.

### 2.6. Validation of Differential Gene Expression by qRT-PCR

The ensure the reliability of transcriptomic data, 12 DEGs were selected for qRT-PCR validation. The accession number for these 12 DEGs were obtained through alignment with *Pyrus bretschneideri* (taxid:225117) in NCBI (Appendix A). Specifically, we validated the expression of the following genes: *PySAUR21-like1* (gene38584), *PyAUX15A* (gene38585), *PySAUR21-like2* (gene38587), *PyAUX15A-like1* (gene27027), *PyAUX15A-like2* (newgene29561), *PyBRI1* (gene6361), *PyPP2C24* (gene24075), *PyPYR/PYL4* (gene24413), *PyGID1C-like* (gene22380), *PyGH3.9* (gene34117), *PyAUX/IAA6* (gene31506), *PyAUX/TLP2* (gene25464). In general, the results demonstrated that the expression patterns of RNAs discovered using qRT-PCR are comparable to those discovered using RNA-seq data, strengthening the validity of the DEGs discovered in this work (Figure 6, Appendix A).

### 2.7. Overexpression PyCYCD3 Increased Lateral Branch Formation of Transgenic Arabidopsis

To explore the role of *PyCYCD3* with respect to the induction of lateral branch formation in pear trees, the CDS of the *PyCYCD3* was amplified, and the fusion protein of *pCAMBIA2300-PyCYCD3-GFP* was stably transformed into *Arabidopsis* (Figure 7a and Appendix A). Transgenic *Arabidopsis* was obtained by the Agrobacterium-medicated leaf disc method and PCR identification (Appendix A). *CYCD3-1*, *CYCD3-2*, *CYCD3-3* were selected to explore whether *PyCYCD3* promotes branching in transgenic *Arabidopsis* (Figure 7b). Through the photographic observation and the branching rate analysis of wild type and *PyCYCD3* gene-transferred *Arabidopsis* plants grown for 1 month (Figure 7b,c, Appendix A), it was found that the growth rate and number of branches of transgenic *Arabidopsis* plants were significantly higher than those of wild type *Arabidopsis* plants during the same period. Six transgenic *Arabidopsis* plant lines (*CYCD3-1*, *CYCD3-2*, *CYCD3-3*, *CYCD3-4*, *CYCD3-5*, *CYCD3-6*) were selected to observe how *PyCYCD3* would be expressed by methods of the qRT-PCR (Figure 7d). The qRT-PCR results revealed that the expression level of *PyCYCD3* was significantly up-regulated compared with WT (Figure 7d, Appendix A).

The subcellular localization of *pCAMBIA1300-PyCYCD3-GFP*-transformed tobacco was studied. The results showed that the *PyCYCD3* gene was mainly located in the cytoplasm, while unloaded *pCAMBIA1300-GFP* was expressed in both the cytoplasm and the membrane (Figure 7e). This indicates that *PyCYCD3* gene plays different biological characteristics and functions in different cell physiological processes. Given its primary localization in the cytoplasm, the *PyCYCD3* gene may be related to cell division, mitosis, and other related processes. In three transgenic *Arabidopsis* plant lines, *CYCD3-1*, *CYCD3-2,* and *CYCD3-3*, the ZR content were measured to be 21.57 ng/g, 36.69 ng/g, and 33.99 ng/g, respectively. The GA_3_ contents were 52.91 ng/g, 69.87 ng/g, and 77.12 ng/g, respectively. The contents of IAA were 47.82 ng/g, 54.89 ng/g, and 68.11 ng/g, respectively. These values were significantly higher than those of ZR (10.02 ng/g), GA_3_ (87.68 ng/g), and IAA (17.12 ng/g) in wild *Arabidopsis* (*p* < 0.01) (Figure 7f, Appendix A). Furthermore, the BR/ABA ratios for the transgenic lines (0.31, 0.26, 0.29) were significantly higher than those of wild *Arabidopsis* (0.17) (*p* < 0.05) (Figure 7g, Appendix A). These results indicate that the heterologous expression of *CYCD3* may promote the synthesis of growth-promoting hormones ZR, GA, IAA, and BR and regulate the stability of growth-inhibiting endogenous hormone ABA, thereby modulating plant branch growth.

## 3. Discussion

### 3.1. The Germination, Branching Rate, and Endogenous Hormone Levels in Lateral Buds

The bud-notching technology that was used in this study provides a valuable resource for the exploration of the molecular mechanism of branching regulation in pear trees. Based on the analysis of the transcriptome of specific cells or tissues, RNA-seq is a suitable method to identify the main biological processes or pathways associated with branching [42]. Differences in the RNA-seq data of lateral buds among the S0, S1, and S2 indicated that plant hormone signal transduction was significantly enriched, providing the initial foundation for the present study. The current findings suggest that hormone signal transduction pathways involving IAA, GA, ZR, and CTK facilitate bud germination and promote plant growth, whereas ABA is associated with the induction of bud dormancy and differentiation [19,24]. Furthermore, 6-benzyladenine (6-BA) has the potential to significantly enhance the growth response of tender shoots without fallen buds on 2 or 3-year-old apple trees [9]. In this study, it was found that there was no significant difference in germination rates between the bud-notching and control groups (Figure 2a). The application of bud-notching treatment significantly enhanced the long branches rate of pear trees, as evidenced by the results presented in Figure 2b. Interestingly, the contents of the endogenous plant hormones IAA and BR showed the same increasing trend, and the contents of these two hormones were significantly higher in lateral buds (S2) sprouted by bud-notching treatment compared to the control buds (S0) (Figure 2c). Meanwhile, the effects of other hormones, such as ethylene and salicylic acid, as well as those of uncontrolled environmental factors on branching, were not thoroughly investigated in this study, which had an impact on the results of this study. We will conduct further research in future experiments.

### 3.2. The Genes Related to Phytohormone Signaling in Lateral Bud-Sprouting Regulation

Classical theory posits that apical dominance governs the transport of auxin along the primary stem by mediating the competition among auxin sources and subsequently regulates the activation of lateral buds [43,44]. In apples, the mechanism of auxin transport from axillary buds to stems is integral to the growth of axillary buds, which is considered essential for their development [24]. In this study, we identified the most differentially expressed genes associated with hormone signal transduction within the auxin pathway, specifically including *AUX1*, *AUX/IAA*, *ARF*, *GH3*, and *SUAR*, which are activated by auxin. The expression of *AUX1*, *AUX/IAA*, and *GH3* was found to be up-regulated in the S2 treatment, while *ARF* exhibited up-regulation in both S1 and S2, but was down-regulated in S0 [45,46]. CTK may regulate the growth of apple branches by activating certain factors that indirectly suppress the expression of genes linked to branching [47,48]. Research indicates that type-B *ARRs*, such as *ARR1* and *ARR2*, can activate type-A *ARRs,* such as *ARR6,* in plants [49]. *ARR1*, *ARR2*, and *ARR10* trigger the transcription of *ARR4*, *ARR5*, *ARR6*, and *ARR7* in response to CTK through *AHK4* and/or *CKI1* [50]. The *Arabidopsis* His-Asp phosphorelay network comprises 11 type-B *ARR* members, which are essential transcriptional regulators [51]. The current study demonstrates that the *AHP* and *B-ARR* genes are significantly enriched in the CTK signaling pathway. Specifically, Gene22385 (*AHP*) and Gene9903 (*B-ARR*) exhibited up-regulation in S1 and down-regulation in S2, suggesting that phosphorylated *AHPs* may translocate to the nucleus [52,53]. The finding that type-A ARRs are required for CK-mediated bud outgrowth may explain the increased expression of *MsARR3/5* during bud outgrowth. It is suggested that exogenous GA spraying benefits lateral bud initiation and branch extraction [5,9,47]. This study confirmed this conclusion and identified down-regulated gibberellin genes *GID1* and *TF* in both S1 and S2. Research has demonstrated that *PYR/PYL/RCAR* proteins inhibit *PP2C* phosphatase activity, which is a known inhibitor of the ABA signaling pathway. This suggests that *PYR/PYL/RCAR* and *PP2C* might work together to negatively regulate the ABA signaling pathway [54,55]. *PYR/PYL*, *PP2C*, and *ABF* genes, in the abscisic acid signaling pathway, were identified in this study. *PYR/PYL* (gene24413) was down-regulated in S1 and up-regulated in S2, while *PP2C* (gene12403, gene24075, gene26342) was up-regulated in S1 and down-regulated in S2. *ABF* (gene23064) was up-regulated in both S1 and S2 and down-regulated in S0. Previous research suggests that ABA indirectly promotes bud differentiation rather than playing a direct role [56,57]. Several studies have indicated that ABA increases the vacuole sugar content, leading to enhanced leaf dewatering [57,58,59]. It has also been shown that root pruning and ring cutting may enhance the differentiation process of buds [59]. Decreased IAA content and increased ABA and CTK contents in xylem SAP during flower bud differentiation may lead to increased shoot branching in pear trees through the up-regulation of genes involved in ABA signaling. Similarly, studies in tomato found significantly higher IAA content in axillary buds of *wrky-b*, *bl*, and *pin4* mutant plants than in WT plants [11]. The *CYCD* gene, which plays a crucial role in cell division as an intermediary between endogenous and exogenous developmental signals, is co-regulated by several plant hormones, including auxin, CTK and BR [36,60,61,62]. In this study, *BRI1*, *BSK*, and *CYCD3* genes were found to be enriched in the BR signaling pathway. *BRI1* and *CYCD3* were down-regulated in S1 and up-regulated in S2, while *BSK* was up-regulated in S1 and down-regulated in S2 (Figure 4a). Notably, the up-regulation of *CYCD3* (gene1164) in S2 aligns with findings reported by Tan et al. in 2019. Additionally, the expression of cell proliferation genes such as CYCLIN D3 (*MsCYCD3*) was found to be up-regulated at stage “1” in the more-branching mutant (MB).

### 3.3. The Interactions between mRNAs and miRNAs

MiRNAs are short noncoding RNAs, around 19–24 nucleotides long, that can interact with longer target RNAs [63]. In ‘Huangguan’ pear (*Pyrus bretschneideri* Rehd.), 505 miRNAs were identified, including 143 known and 362 new ones. The interactions between miRNAs and their target genes can be one-to-many or many-to-one, rather than strictly one-to-one, introducing significant regulatory complexity (Figure 4b). This complexity is exemplified by the fact that a single miRNA can regulate multiple genes involved in branching, and conversely, a single gene can be targeted by multiple miRNAs. This intricate network underscores the elaborate control mechanisms governing branching in young pear plants. Specifically, we found ten miRNAs targeting a single gene (*ABF*) related to pear shoot branching, namely aly-miR157d-3p, cem-miR166g, atr-miR166b, gma-miR166m, bdi-miR166e-3p, hbr-miR166a, cpa-miR166e, aly-miR166a-3p, mes-miR166i, cis-miR166a. In addition, only one miRNA was identified to target at least two pear shoot branching-related genes: cpa-miR166e (*GH3* and *ABF*). One miRNA was identified to target each of the pear shoot branching-related genes:cpa-miR166e (*GH3*), gma-miR390e (*SAUR*), gma-miR162a (*SAUR*), hbr-miR156 (*PP2C*), aly-miR390a-5p (*BIR1*), and cme-miR396e (*CYCD3*). *CYCD3;1*, a key player in cell proliferation and differentiation during plant development, is expressed in growing tissues, such as meristems and developing leaves [64]. *BIR1*, another important gene, prevents cell death, and its inactivation in yeast leads to cell division defects [65]. Our study revealed that the up-regulation of *BIR1* and *CYCD3* is crucial for shoot branching in pears. In summary, cap-miR166e was up-regulated, while *GH3* (gene11049) was down-regulated in S2 compared to S0. Conversely, gma-miR162a, hbr-miR156, aly-miR390a, and cem-miR396e were down-regulated in S2, leading to the up-regulation of their respective target genes: SAUR (gene27024), *PP2C* (gene12403), *BIR1* (gene6361), and *CYCD3* (gene1164).

### 3.4. The Over-Expression of PyCYCD3 in Arabidopsis

The influence mechanisms of hormones such as IAA, CTK, GA on lateral bud sprouting have been extensively studied [19,24]. However, the role of BR in this process remains less well understood. Consequently, our research has concentrated on the *CYCD3* gene within the BR signaling pathway. The BES1/BZR1-ARF complex supports the longstanding reported synergistic interaction between auxin and BRs, implicated in both elevated gene expression and extended hypocotyl growth [66,67]. It is speculated that *ARF* and *CYCD3* may have similar synergy in promoting lateral bud germination. The overexpression of *PtoCYCD3* promoted the cell cycle G1/S transition of perennial poplars, thus accelerating the cell division of poplar meristem, such as apical buds, axillary buds, and stem cambium; it also accelerated plant vegetative growth and produced obvious branches [68]. *OsCYCD3;1* may maintain axillary meristem to promote branching in rice, possibly by regulating cell division [69]. In poplar, the upregulation of *AHP* and *CYCD3* expression levels is closely related to branch formation [70]. This is consistent with the results of this study. Transgenic *Arabidopsis* leaf explants overexpressing *CycD3* can initiate and maintain cell division in the absence of CTK, suggesting that CTKs activate cell division through induction of *CycD3* at the G1/S transition [62]. This evidence indicates that plant D-type cyclins may function as mediators of internal and environmental stimuli to drive cell division. Our study indicated that the heterologous over-expression of *PyCYCD3* has a positive regulatory effect on the increased *Arabidopsis* growth and branching number (Figure 7b,c). Therefore, up-regulated *PyCYCD3* may play a crucial role in accumulating auxin and BR, which function as mediators of bud-notching stimuli to drive cell division and regulate lateral branching processes in pear trees.

## 4. Materials and Methods

### 4.1. Plant Materials and Sample Collection

Plant materials were obtained from 2-year-old grafted saplings of the pear variety ‘Huangguan’ (*Pyrus bretschneideri* Rehd.), cultivated at the National Modern Agricultural Pear Industrial Technology Experimental Station of Tiaoshan Group (Jingtai, Gansu Province, China). It is located at 103°59′~104°05′ E, 37°07′~37°09′ N, with an annual average temperature of 8.25 °C, an average frost-free period of 141 days, and an annual average precipitation of 183.1 mm. The main soil types are calcareous soil and cultivated soil. On April 5th, before the bud germination of pear seedlings, seedlings with consistent height and vigor standards were selected for budding treatment at a designated location. The specific operation method involves incising 0.4 cm above the bud body between 30–60 cm below the pear tree’s top bud, with the incision being half the stem’s circumference. The control group pear trees were not subjected to budding treatment. On the 15th day after budding, the first sample (S0) was collected from the first bud below 30 cm from the central stem to the top bud of the control tree, which was not budded and generally cannot grow into long branches. The second sample (S1) was taken from the first bud and tender shoot closest to the top bud of the control tree, which was not budded and can grow into long branches. The third sample (S2) was gathered from the first bud and tender shoot below 30 cm from the central stem to the top bud of the budded tree, which can grow into long branches after budding. All samples collected at each position had at least 30 buds and tender shoots mixed into one sample, with three biological replicates set for both the treatment and control groups [9,71].

### 4.2. Measurement of Emergence Rate, Branching Rate, and Hormone Levels in Lateral Buds

After the leaves fell in autumn, the number of buds and the growth length of the germination rate, the peripheral extended branch length and branch analogy (long branch > 15 cm, middle branch 5~15 cm, short branch < 5 cm) were assessed. The quantities of long, medium, and short branches, along with the average growth length of all branches were independently measured for both untreated and bud-notched pear trees.

High-performance liquid chromatography was used to quantify the contents of indole acetic acid (IAA), zeatin riboside (ZR), abscisic acid (ABA), and gibberellic acid (GA_3_) in 0.2 g of freshly lateral buds, collected at 15 days post-sampling, in accordance with previously described methods [19,72]. For the quantification of the brassinosteroid (BR) content, 1 g of freshly collected lateral buds at the same 15-day time point was required. Each sample set for both hormone types was composed of three biological replicates to ensure the reliability and reproducibility of the results.

### 4.3. RNA Preparation and Transcriptome Sequencing

Total RNA was isolated utilizing the Trizol Reagent (Invitrogen, Carlsbad, CA, USA) in accordance with the manufacturer’s guidelines. The NanoDrop 2000 Spectrophotometer was used to measure RNA concentration and purity (Thermo Fisher Scientific, Wilmington, DE). The RNA Nano 6000 Assay Kit of the Agilent Bioanalyzer 2100 System was implemented to assess RNA integrity (Agilent Technologies, Santa Clara, CA, USA). Following a manufacturer’s recommendations, sequencing libraries were generated using NEBNextR UltraTM Directional RNA Library Prep Kit for IlluminaR (NEB, USA) and the small RNA Sample Library Prep Kit for IlluminaR (NEB, Ipswich, MA, USA), and index codes were added to assign sequences to each sample. Biomarker Technologies Corporation addressed library preparation and sequencing (Beijing, China). The Agilent Bioanalyzer 2100 and qPCR were employed to analyze library quality. The RNA-seq data were submitted to the NCBI SAR database under the accession number PRJNA933274 (https://www.ncbi.nlm.nih.gov/bioproject/?term=PRJNA933274 (accessed on 9 February 2023)).

### 4.4. Differential Expressed Genes (DEGs) Analysis

Differential expression analysis was performed using the DESeq (2010) R package. The resulting FDR (false discovery rate) was adjusted using the PPDE (posterior probability of being DE). An FDR < 0.05 and ǀlog_2_(FoldChange)ǀ ≥ 1 were set as the threshold for significantly differential expression. Gene function was annotated based on the KEGG database (Kyoto Encyclopedia of Genes and Genomes). KOBAS 3.0 software (http://kobas.cbi.pku.edu.cn/kobas3) was used to test the statistical enrichment of DEGs in the KEGG pathway [73].

### 4.5. Quantitative Real-Time PCR (qRT-PCR) Validation

A total of twelve DEGs with significant expression patterns of the KEGG pathway were selected for qRT-PCR verification. Purification and reverse transcription of RNA were used PrimeScript™ RT reagent Kit with gDNA Eraser (TAKARA). Primers were designed by online software Primer3 input (version 4.0, http://primer3.ut.ee/ (accessed on 28 March 2023)), and the sequences are shown in Appendix A. The qRT-PCR was conducted on CFX96 Touch™ Real-Time PCR Detection System (Bio-Rad, Hercules, CA, USA) by using SYBR^®^Premix ExTaq™ II. The data were analyzed by 2^−∆∆Ct^.

### 4.6. Overexpression PyCYCD3 in Arabidopsis and Subcellular Localization Analysis

The genomic sequences of the CDS of the *PyCYCD3* gene were amplified from pear genomic DNA by the use of primers (*F*:AGAACACGGGGGACGAGCTCATGGGTGGTTGGGCAATTGCAG; *R*:ACCATGGTGTCGACTCTAGACTCCCAAATCCCAACCTCCATG), after which the sequences were cloned into *pCAMBIA2300-GFP* vectors and double-digested using SacI and XbaI, resulting in *pCAMBIA2300-PyCYCD3-GFP* constructs. The inverted trifoliate and inverted quadrifoliate leaves of Ben’s tobacco were selected for transient transformation of *PyCYCD3* to determine the subcellular localization of the gene. Subsequently, the *PyCYCD3* gene was transfected into *Arabidopsis* Columbia using the floc dipping method, and the positive plants were screened on MS plates containing 50 mg/L Kan and 20 mg/L tauramycin, after which the transgenes obtained from the screening were identified by PCR assay, and the T3 generation of pure transgenic *Arabidopsis* plants was obtained after two screenings. Finally, the number of transgenic *Arabidopsis* plants branching, the relative expression of *PyCYCD3* gene (*PyCYCD3-F*: CGCCGTTTCGGGATTAACTA; *PyCYCD3-R*: GAGGTAGGAGTGAGGTGATTTG. *Actin-F*: GCCGACAGAATGAGCAAAGAG; *Actin-R*: AGGTACTGAGGGAGGCCAAGA), and its endogenous hormone content were determined.

### 4.7. Data Analysis

A total of thirty plants of both transgenic and wild-type *Arabidopsis* were selected for observation of lateral shoot growth following a period of shoot extraction. The data obtained were subjected to statistical analysis using the independent samples *t*-test feature in SPSS 22.0 software for comparison purposes. Two-tailed *t*-tests were employed to identify significant differences in the sample results. Additionally, Microsoft Excel 2019 was utilized for the statistical analysis of the raw data, while graphing was performed using Origin 9.0.

## 5. Conclusions

In conclusion, this study involved sequencing the transcriptome of bud-notched and control buds in pear (*Pyrus bretschneideri* Rehd.), focusing on plant hormone signal transduction pathways and analyzing the mRNA-miRNA interactions. A total of 15 miRNA-mRNA pairs were investigated, revealing both positive and negative correlations. Subsequently, *PyCYCD3* was chosen for transformation into wild *Arabidopsis* plants, resulting in the generation of transgenic *Arabidopsis* plants. The analysis of branching patterns, endogenous hormone levels, and the relative expression of *PyCYCD3* in these transgenic *Arabidopsis* plants suggests that *PyCYCD3* may potentially play a significant role in promoting lateral branches in pear trees through bud-notching techniques. The results demonstrated that BR signaling may integrate with auxin signaling pathways and up-regulate the *PyCYCD3* (down-regulate cem-miR396e) to control the formation of lateral branches by promoting lateral bud sprouting.

## Figures and Tables

**Figure 1 plants-13-02928-f001:**
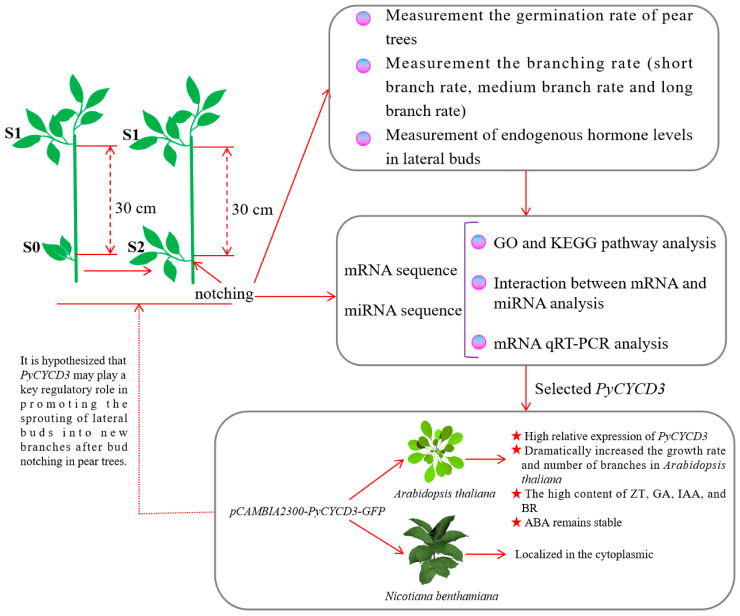
Simplified schematic diagram of the main research in this study.

**Figure 2 plants-13-02928-f002:**
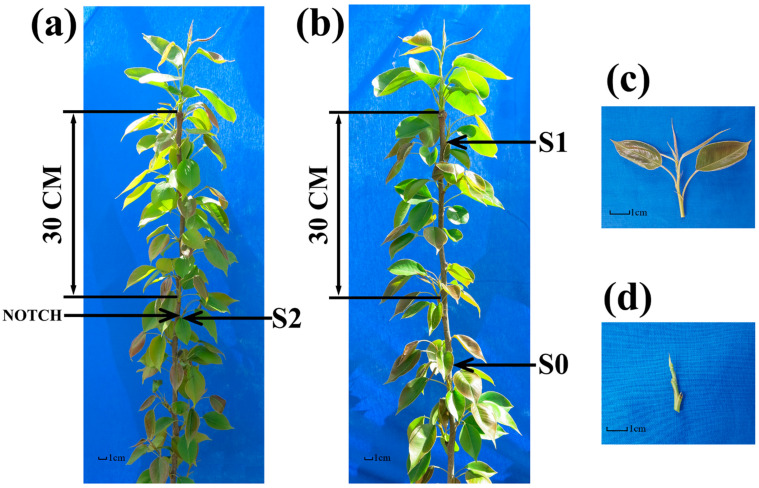
Schematic diagram of sample collection. (**a**) S2 represents the third bud sample, which consists of lateral buds that can sprout normally after bud-notching. (**b**) S0 represents the first bud sample, which consists of lateral buds that can sprout normally after bud-notching. S1 represents the second bud sample, which consists of lateral buds that can sprout normally after bud-notching. (**c**) Illustrates either S1 or S2. (**d**) Represents S0.

**Figure 3 plants-13-02928-f003:**
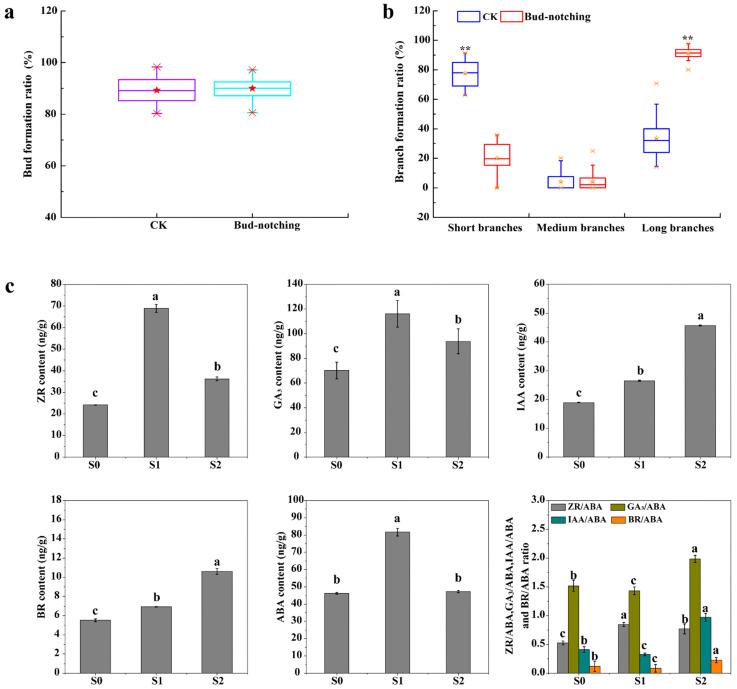
Germination rate, branching rate, and endogenous hormone levels in lateral buds. (**a**) Sprouting rate of pear trees after bud-notching treatment. CK, control. (**b**) Comparison of branch formation rate between control and bud-notching treatment groups. Note: the test in Figure 3a,b is an independent samples *t*-test; *p* < 0.01 indicates a highly significant difference (**). (**c**) The endogenous content and ratio of hormones. Note: the hormone contents and the ratios of the hormone contents and ratios are the means of three independent samples. a, b, and c indicate a significant difference at the *p* < 0.05 level.

**Figure 4 plants-13-02928-f004:**
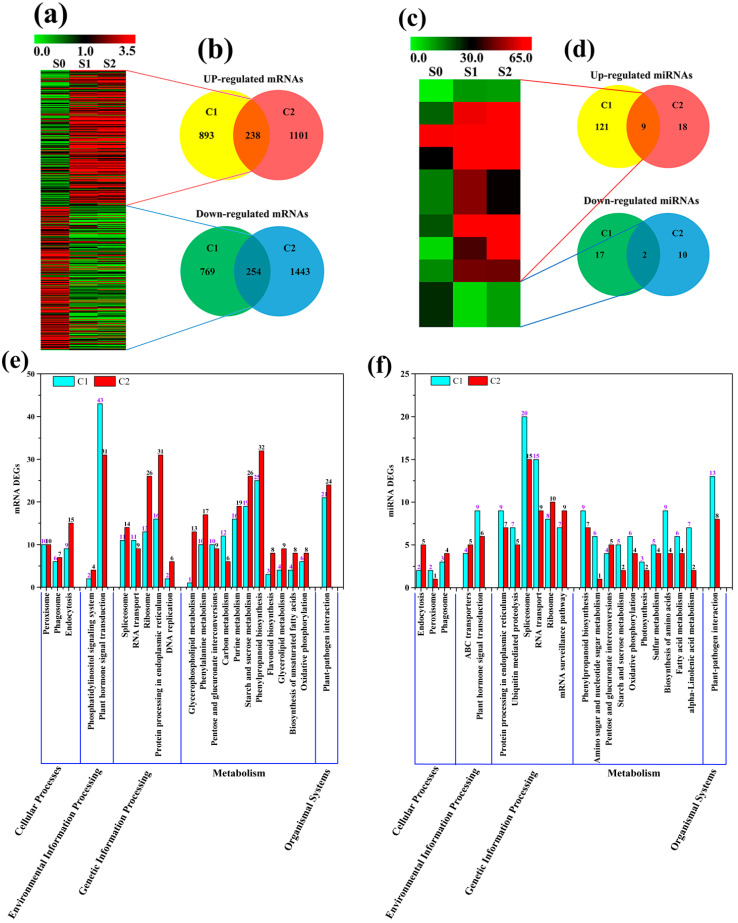
Differentially expressed mRNAs and miRNAs in both confrontations, C1 and C2. (**a**) The cluster heatmap of 492 DEGs at both confrontation C1 and C2. (**b**) The number of up-regulated (yellow, red) and down-regulated (green, blue) DEGs detected between C1 and C2. (**c**) The cluster heatmap of 11 DEGs at both confrontation C1 and C2. (**d**) The number of up-regulated (yellow, red) and down-regulated (green, blue) miRNAs detected between C1 and C2. (**e**) Pathway analysis of mRNA DEGs based on the KEGG database C1 and C2. (**f**) Pathway analysis of miRNA DEMs based on the KEGG database at C1 and C2. C1 is the ratio of S0_ vs. _S1, C1 is the ratio of S0_ vs. _S2.

**Figure 5 plants-13-02928-f005:**
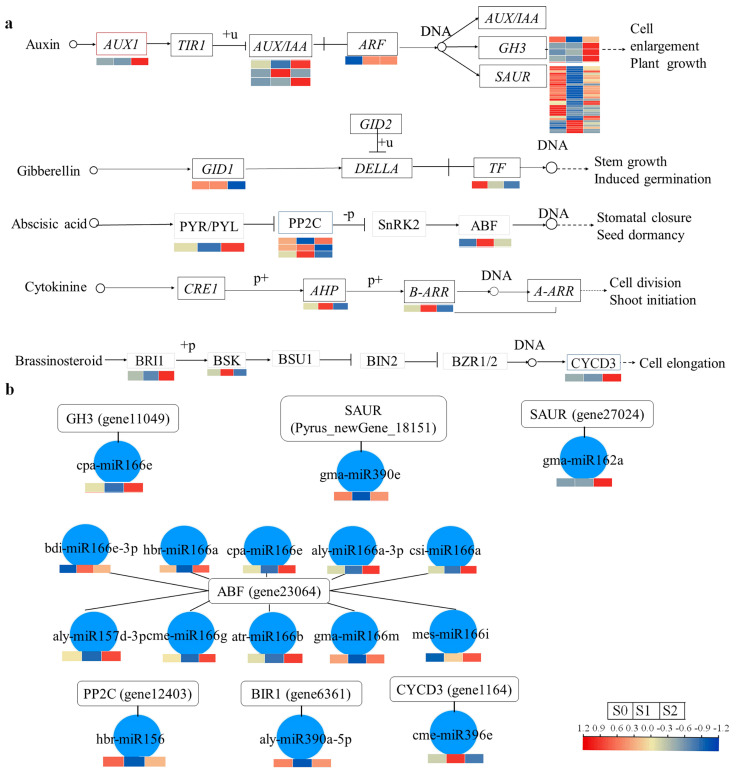
Pathway analysis of mRNA DEGs and miRNA DEGs based on the KEGG database. (**a**) Pathway analysis of mRNA DEGs based on the KEGG database for Auxin, Gibberellin, Abscisic acid, Cytokinine, and Brassinosteroid. +p, phosphorylation; −p, dephosphorylation; +u, ubiquitination; p+, in a phosphorylated state. (**b**) A regulatory network of miRNA-mRNA related to pear shoot branching-related genes.

**Figure 6 plants-13-02928-f006:**
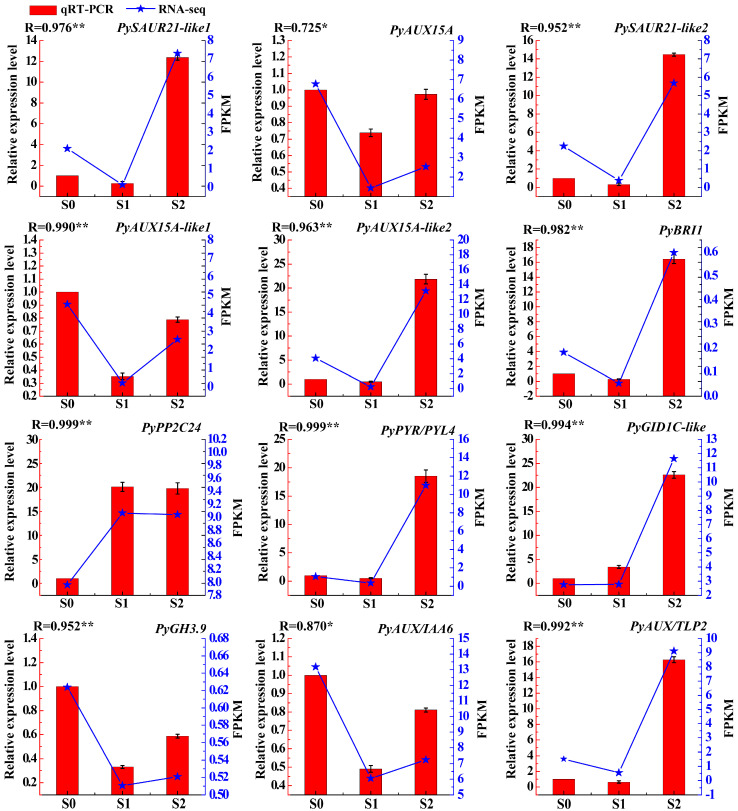
Real-time PCR validation of 12 DEGs in mRNA pairs. The *x*-axis represents RNA names, with the left *y*-axis represents the relative expression levels, and the right *y*-axis represents FPKM values. The red bars represent data yielded by qRT-PCR, while the blue star points represent data obtained by RNA sequencing; ‘r’ represents the Pearson correlation coefficient. ‘*’, correlation is significant at the 0.05 level. ‘**’, correlation is significant at the 0.01 level.

**Figure 7 plants-13-02928-f007:**
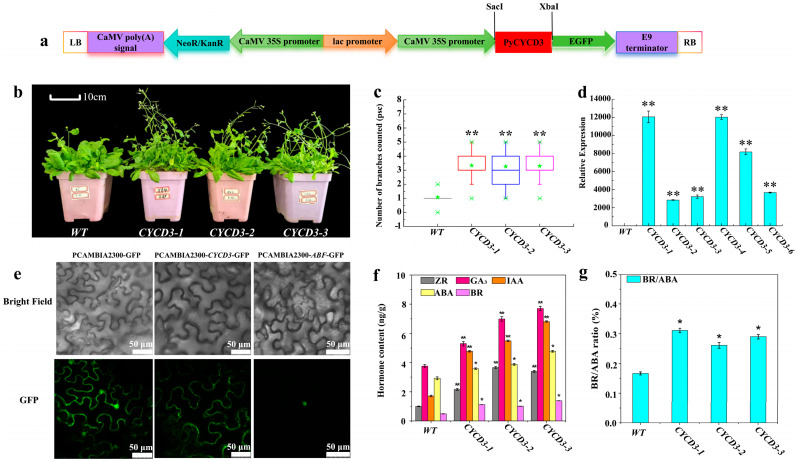
Overexpression *PyCYCD3* increased lateral branch formation of transgenic *Arabidopsis*. (**a**) Schematic of the *pCAMBIA2300-PyCYCD3-GFP* fusion construct. (**b**) The phenotypic difference of WT and transgenic *Arabidopsis thaliana*. (**c**) Comparison of the number of outgrowths in WT and transgenic *Arabidopsis thaliana*. (**d**) The relative expression levels of *PyCYCD3* in WT and transgenic lines. (**e**) Subcellular localization of *PyCYCD3* gene in tobacco leaves. Bright field: indicates bright field of view; GFP: indicates target protein GFP signal. Bar = 50 μm. (**f**) The content of ZR, GA_3_, IAA, ABA, and BR in branches of WT and transgenic *Arabidopsis*. (**g**) The BR/ABA of hormones in branches of WT and transgenic *Arabidopsis*. The test in figure is an independent samples *t*-test; *p* < 0.05 indicates a significant difference (*), and *p* < 0.01 indicates a highly significant difference (**).

## Data Availability

The data presented in this study are available on request from the corresponding author. The data are not publicly available due to the regulatory restrictions of the funding organization.

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
