# Peer review of "Integration of mRNA-miRNA Reveals the Possible Role of PyCYCD3 in Increasing Branches Through Bud-Notching in Pear (Pyrus bretschneideri Rehd.)"

_plants, 2024, doi:10.3390/plants13202928_

Round 1
Reviewer 1 Report
Comments and Suggestions for Authors
Reviewer’s comments
In the manuscript of An et al. entitled “Integration of mRNA-miRNA Revealed the Possible Role of 2 PyCYCD3 may Increase Branches through Bud-Notching in 3 Pear (Pyrus bretschneideri Rehd.)” the authors following a series of experimental approaches such as measurements of germination and branching rates, quantification of hormone levels in lateral buds of pear trees as well as transcriptomics analysis after bud notching and tried to shed light on the activation of hormonal signaling pathways and molecular mechanisms to promote healthier growth and early yield. The work at the experimental level is well-designed and performed, however, I have some difficulties following how the results are presented. In this context, please find below my suggestions:
1. I think the order of the figures should change. For example, Figure 6 should move at the beginning of the results because it refers to the sample collection, something significant for the reader to understand the following analyses. Furthermore, Figure 7 should be presented earlier in the text as the experimental workflow followed in this work.
2. In lines 115-118, I think there is an inconsistency between what you write and the results you present. For example, you mention that ABA concertation does not show any significant difference among S0, S1 and S2, but in the presented results there is significant difference in S1. The same stands for the ratio of BR/ABA and auxin/ABA. Please check.
3. In figure 3. Please correlate miRNA expression with the expression of genes that control.
Minor comments.
1. In lines 10-11, the sentence needs English editing.
2. In line 56 , write the full name of CTK.
3. In line 97 you write “ 60 cm radius.” Are you sure about the number?
4. In the Figure 1 legend, please write that Ck is the control and what is the meaning of S0, S1, and S2. In addition, change hormo0nes” to hormones
5. In Figure 3, the meaning of + and -p should be described.
6. In the legend of Figure 3, please mention that C1 is the ratio of S0 vs S1, and C2 is the ratio of S0 vs S2.
7. In Figure 5g. In the Y-axis “ration of BR/ABA” should change to ratio.
8. In line 141, change “regulated” to regulated.
Comments on the Quality of English LanguageNo particular issues with the quality of English language only some spelling mistakes were detected.
Author Response
Authors’ response to reviewers’ comments
Dear Editors and Reviewers,
Thank you for your letter and for the comments concerning our manuscript titled " Integration of mRNA-miRNA Revealed the Possible Role of PyCYCD3 may Increase Branches through Bud-Notching in Pear (Pyrus bretschneideri Rehd.) (plants-3211069). The comments have helped us improve our manuscript and have been valuable for our future study. We have studied the comments carefully and have applied corrections, which we hope would meet your approval. The main corrections in the paper and the response to the comments are as follows:
Re-Reviewers' comments:
Re-Reviewer #1:
Comment1: I think the order of the figures should change. For example, Figure 6 should move at the beginning of the results because it refers to the sample collection, something significant for the reader to understand the following analyses. Furthermore, Figure 7 should be presented earlier in the text as the experimental workflow followed in this work.
Response: Thank you very much for your valuable suggestions. We have moved the Figure7 to "Introduction" and corrected to the Figure1, moved Figure6 to the beginning of the results and corrected to the Figure2, the order of the following figures has been modified in turn and is marked in red.
Comment2: In lines 115-118, I think there is an inconsistency between what you write and the results you present. For example, you mention that ABA concertation does not show any significant difference among S0, S1 and S2, but in the presented results there is significant difference in S1. The same stands for the ratio of BR/ABA and auxin/ABA. Please check.
Response: Thank you very much for your valuable suggestions. In lines 115-118, we have correctecd the "ABA concertation does not show any significant difference among S0, S1 and S2" to the "while the concentration of ABA did not exhibit a significant difference among S0, and S2, but the levels of ABA in S1 was significantly elevated compared with S0 (P<0.05). "(line145-147)
Comment3: In figure 3. Please correlate miRNA expression with the expression of genes that control.
Response: The Figure 3 have been corrected to the Figure5, the analysis shows that the relationship between genes and miRNAs is primarily negative, as described in lines 258-261.
Minor comments.
Comment1: In lines 10-11, the sentence needs English editing.
Response: We have edited the sentence of line 10-11 by English.
Comment2: In line 56 , write the full name of CTK.
Response: In line 56, we have write the full names of the CTK, cytokinin.
Comment3: In line 97 you write “ 60 cm radius.” Are you sure about the number?
Response: In line 97 you write “ 60 cm radius.” We sure about the number.
Comment4:In the Figure 1 legend, please write that Ck is the control and what is the meaning of S0, S1, and S2. In addition, change hormo0nes” to hormones
Response: The Figure1 have been corrected to the Figure3, and we have write CK is control in the figure3 legend (line155). The meaning of S0, S1, and S2 as show in result 2.1. Samples treatment and collection (line 102-116). In addition, we have changed the "hormo0nes” to "hormones".
Comment5: In Figure 3, the meaning of + and -p should be described.
Response: In Figure 3 have been corrected to the Figure5, the meaning of +p, phosphorylationï¼›-p, dephosphorylationï¼›+u, ubiquitination; p+, in a phosphorylated state. (line241-242)
Comment6: In the legend of Figure 3, please mention that C1 is the ratio of S0 vs S1, and C2 is the ratio of S0 vs S2.
Response: The mention Figure 3 that is the Figure 2, which have been corrected to the Figure 4. In the legend of Figure 4, we have added the note of the C1 is the ratio of S0 vs S1, and C2 is the ratio of S0 vs S2. (line211)
Comment7: In Figure 5g. In the Y-axis “ration of BR/ABA” should change to ratio.
Response: Figure 5 have been corrected to the Figure 7, In the Y-axis "ration of BR/ABA" have been changed to the ratio.
Coment8: In line 141, change “regulated” to regulated.
Response: In line 172-173, we have changed “regulaed” to regulated.
Reviewer 2 Report
Comments and Suggestions for Authors
Dear Authors,
The manuscript titled "Integration of mRNA-miRNA Revealed the Possible Role of PyCYCD3 in Increasing Branches through Bud-Notching in Pear (Pyrus bretschneideri Rehd.)" offers a valuable contribution to the field by exploring the molecular underpinnings of bud-notching in pear trees. The focus on PyCYCD3 is especially insightful, as it delves into the gene's role in hormonal signaling pathways that influence branching, an area that could have practical implications for pear cultivation. The use of RNA-seq analysis and miRNA-mRNA interaction studies adds depth to the research. However, there are several areas that require improvement to enhance the clarity and scientific rigor of your work. Below are my recommendations for revisions:
One critical issue that stands out is that the references cited in the manuscript are missing recent studies from 2022 to 2024. Given the rapid advances in plant physiology and molecular biology, I recommend that you update your literature review and citations to include more recent publications from the last two years. This will ensure that your study is grounded in the latest research and aligns with current developments in the field.
While the introduction provides a good general overview, it does not sufficiently highlight the research gap that your study aims to address. It would be beneficial to explicitly state what remains unclear in the field of hormonal regulation related to bud-notching in pear trees. This could include gaps in our understanding of the molecular interactions driving branching in pears or how PyCYCD3 specifically contributes to these pathways. Clearly stating these gaps will not only strengthen the novelty of your research but also better define the value of your contribution to the broader scientific community.
Environmental factors like temperature, light conditions, and soil composition play a crucial role in plant hormone regulation and responses. However, these aspects are not sufficiently detailed in the current manuscript. To strengthen the reproducibility and scientific validity of your findings, I recommend that you provide more information regarding the environmental conditions under which the pear trees were grown. Moreover, while the hormonal assays you conducted are relevant, there is no clear justification provided for the exclusion of important hormones such as ethylene and salicylic acid. Adding this justification, along with the reasoning behind your specific sampling time points, would help clarify your experimental design and make the study more comprehensible to readers.
The results section presents detailed statistical comparisons, but it feels overly dense, which may make it harder for readers to quickly grasp the key findings. To improve the flow and accessibility of your results, I suggest summarizing the most important statistical data in tables or figures. This will help readers focus on the most critical findings without being overwhelmed by dense text. Simplifying the way you present your results will enhance the overall readability and impact of this section.
I noticed that Figure 5B and Figure 6 are missing the bar size for scale, which is critical for interpreting the images accurately. Adding the scale bar to these figures will improve their clarity and ensure the data is correctly understood by the readers. Please include appropriate scale bars and mention them in the figure legends.
The discussion does a good job of interpreting your results, but I believe it could be improved by linking your findings more directly to the broader literature on hormonal regulation and practical applications. For instance, how might your findings on PyCYCD3 influence pear breeding programs or horticultural practices aimed at enhancing branching and yield? It would also be helpful to acknowledge potential limitations in your study, such as the exclusion of key hormones like ethylene or the influence of environmental factors that were not controlled for. Recognizing these limitations will add depth to your discussion and provide a balanced interpretation of the results.
Author Response
Authors’ response to reviewers’ comments
Dear Editors and Reviewers,
Thank you for your letter and for the comments concerning our manuscript titled " Integration of mRNA-miRNA Revealed the Possible Role of PyCYCD3 may Increase Branches through Bud-Notching in Pear (Pyrus bretschneideri Rehd.) (plants-3211069). The comments have helped us improve our manuscript and have been valuable for our future study. We have studied the comments carefully and have applied corrections, which we hope would meet your approval. The main corrections in the paper and the response to the comments are as follows:
Re-Reviewers' comments:
Re-Reviewer #2:
Comment: One critical issue that stands out is that the references cited in the manuscript are missing recent studies from 2022 to 2024. Given the rapid advances in plant physiology and molecular biology, I recommend that you update your literature review and citations to include more recent publications from the last two years. This will ensure that your study is grounded in the latest research and aligns with current developments in the field.
Response: Thank you very much for your valuable suggestions. Based on your comments, we added the four references. The related references are shown below:
- Zhang, L.L.; Fang, W.M.; Chen, F.D.; Song, A.P. The Role of Transcription Factors in the Regulation of Plant Shoot Branching. Plants. 2022, 11(15):1997.
- Yang, Y.R.; Hu, Y.L.; Li, P.; Hancock, J.T.; Hu, X.Y. Research Progress and Application of Plant Branching. Phyton-International Journal of Experimental Botany. 2023, 92(3), 680-689.
- Gonin, M.; Salas-González, I.; Gopaulchana, D.; Frene, J.P.; Rodend S.; Poel, B.V.D.; Salta, D.E.; Castrillo, G. Plant microbiota controls an alternative root branching regulatory mechanism in plants. Proceedings of the National Academy of Sciences United States of America. 2023, 120(5), e2301054120.
- Yang, H.H.; Zhou, K.; Wu, Q.F.; Jia, X.Y.; Wang, H.X.; Yang, W.H.; Lin, L.H.; Hu, X.M.; Pan, B.Q.; Li, P.; Huang, T.T.; Xu, X.Y.; Li, J.F.; Jiang, J.B.; Du, M.M. The tomato WRKY-B transcription factor modulates lateral branching by targeting BLIND, PIN4, and IAA15. Horticulture Research. 2024, 11, uhae193.
Comment: While the introduction provides a good general overview, it does not sufficiently highlight the research gap that your study aims to address. It would be beneficial to explicitly state what remains unclear in the field of hormonal regulation related to bud-notching in pear trees. This could include gaps in our understanding of the molecular interactions driving branching in pears or how PyCYCD3 specifically contributes to these pathways. Clearly stating these gaps will not only strengthen the novelty of your research but also better define the value of your contribution to the broader scientific community.
Response: Thank you very much for your valuable suggestions. Based on your comments, we added the “Although some previous work has been done on pear budburst, the mechanisms of how its pear budburst-related hormones and associated genes are regulated remain unclear.” in lines 84-86.
Comment: Environmental factors like temperature, light conditions, and soil composition play a crucial role in plant hormone regulation and responses. However, these aspects are not sufficiently detailed in the current manuscript. To strengthen the reproducibility and scientific validity of your findings, I recommend that you provide more information regarding the environmental conditions under which the pear trees were grown. Moreover, while the hormonal assays you conducted are relevant, there is no clear justification provided for the exclusion of important hormones such as ethylene and salicylic acid. Adding this justification, along with the reasoning behind your specific sampling time points, would help clarify your experimental design and make the study more comprehensible to readers.
Response: Thank you very much for your valuable suggestions. As you may be concerned. We added the contents and conferences in lines 449-466.
Comment: The results section presents detailed statistical comparisons, but it feels overly dense, which may make it harder for readers to quickly grasp the key findings. To improve the flow and accessibility of your results, I suggest summarizing the most important statistical data in tables or figures. This will help readers focus on the most critical findings without being overwhelmed by dense text. Simplifying the way you present your results will enhance the overall readability and impact of this section.
Response: Thank you very much for your valuable suggestions. As you are concerned, we have taken this into account. Nevertheless, in the attachment, we have added plenty of schedules to explain the picture related to it. Therefore, we have not added any more relevant tables in the manuscript. Once again, many thanks to the reviewers for their comments.
Comment: I noticed that Figure 5B and Figure 6 are missing the bar size for scale, which is critical for interpreting the images accurately. Adding the scale bar to these figures will improve their clarity and ensure the data is correctly understood by the readers. Please include appropriate scale bars and mention them in the figure legends.
Response: Thank you very much for your valuable suggestions. We have added the ruler to the original Figure 7B (Figure 5B) and the original Figure 6 (Figure 2).
Comment: The discussion does a good job of interpreting your results, but I believe it could be improved by linking your findings more directly to the broader literature on hormonal regulation and practical applications. For instance, how might your findings on PyCYCD3 influence pear breeding programs or horticultural practices aimed at enhancing branching and yield? It would also be helpful to acknowledge potential limitations in your study, such as the exclusion of key hormones like ethylene or the influence of environmental factors that were not controlled for. Recognizing these limitations will add depth to your discussion and provide a balanced interpretation of the results.
Response: Thank you very much for your valuable suggestions. Sections 3.1 and 3.2 of the manuscript extensively discuss the effects of hormones on branching. In addition, we have only included references [9] and [11] in the discussions on lines 333-335 and 384-386. As you mentioned, we have referenced some literature (68, 69, 70) and added content related to the application of CYCD3 on plant branching in the highlighted part of section 3.4 of the manuscript. Regarding the effects of other hormones on the experiment, I would like to point out that there is scarce research on the effects of other hormones on plant branching such as ethylene and salicylic acid in previous studies, and our transcriptome analysis did not reveal enriched RNA-seq analysis results of the ethylene signaling pathway. Therefore, we did not study other hormones. Furthermore, we added the contents "In addition, this study did not delve into the effects of other hormones such as ethylene and salicylic acid as well as uncontrolled environmental factors on branching, which impacted the research results. We will conduct further research in future experiments" in lines 341-345.
Round 2
Reviewer 2 Report
Comments and Suggestions for Authors
Dear Authors,
Thank you for your diligent efforts in revising your manuscript, "Integration of mRNA-miRNA Revealed the Possible Role of PyCYCD3 may Increase Branches through Bud-Notching in Pear (Pyrus bretschneideri Rehd.)". You have addressed the reviewer comments thoroughly, and your revisions have notably improved the quality and clarity of the manuscript, and I believe it is now suitable for publication.
Best regards,